# A Review of Clean Energy Exploitation for Railway Transportation Systems and Its Enlightenment to China

**Jing Teng** [1] , **Longkai Li** [1] , **Yajun Jiang** [1] **and Ruifeng Shi** [1,2,*]

1. School of Control and Computer Engineering, North China Electric Power University, Beijing 102206, China
2. China Institute of Energy and Transportation Integrated Development, North China Electric Power University, Beijing 102206, China
* Correspondence: shi.ruifeng@ncepu.edu.cn

**Abstract:** According to the International Energy Agency (IEA), China's rail system will become fully electrified by 2050. However, in some remote areas with a weak power grid connection, the promise of an electrified railway will be hard to achieve. By replacing conventional fuels with clean and environmentally-friendly energy, overall carbon emissions would be significantly reduced, contributing to the fulfillment of the carbon-neutral commitment. This study reviews clean energy exploitation in the railway transportation system and the distribution of renewable energy sources along the railway lines of China. The evaluation results show that China has huge energy potential. In terms of photovoltaics alone, the annual power generation of China's high-speed railway is about 170 TWh, meaning that the energy self-consistency rate for high-speed railway can reach 284.84%. Efficient exploitation of clean energy sources for China's railway transportation system would effectively mitigate anxieties surrounding energy shortages.

**Keywords:** clean energy; low-carbon transportation operations; railway transportation

## 1. Introduction

The energy and transportation industries form the core of China's economic system. Developing energy production and transportation industries that cooperate will ensure the efficient operation of the economy and continuous productivity improvements. The integration of the two industries plays an irreplaceable role in achieving the goal of carbon neutrality by 2060 [1,2].

Among the various transportation modes, the railway is widely driven by electric energy; thus, it is internationally recognized as a low-carbon transportation choice. Figure 1 shows the greenhouse gas (GHG) intensities of the motorized passenger transport modes, as reported by the International Energy Agency (IEA). As can be seen from Figure 1, the railway has the lowest average $gCO_2$-eq./passenger-km.

The actual proportion of low-carbon and zero-carbon fuels in the passenger rail system in 2020 and the predicted proportion in 2030 according to the forecast of IEA are shown in Table 1. It is projected that electricity and hydrogen will cover the energy needs of the passenger rail system by 2030.

Reducing the fossil fuel consumption of the railway industry plays a pivotal role in achieving carbon neutrality. However, access to electrical railways is extremely limited in some remote areas with weak grids, which makes it difficult to realize net-zero emissions as planned. We focus on exploiting clean energy along the railway lines to overcome these challenges. The integrated development of clean energy and railway transportation is expected to greatly improve the energy exploitation structure and inject new vitality into the railway industrial transformation.

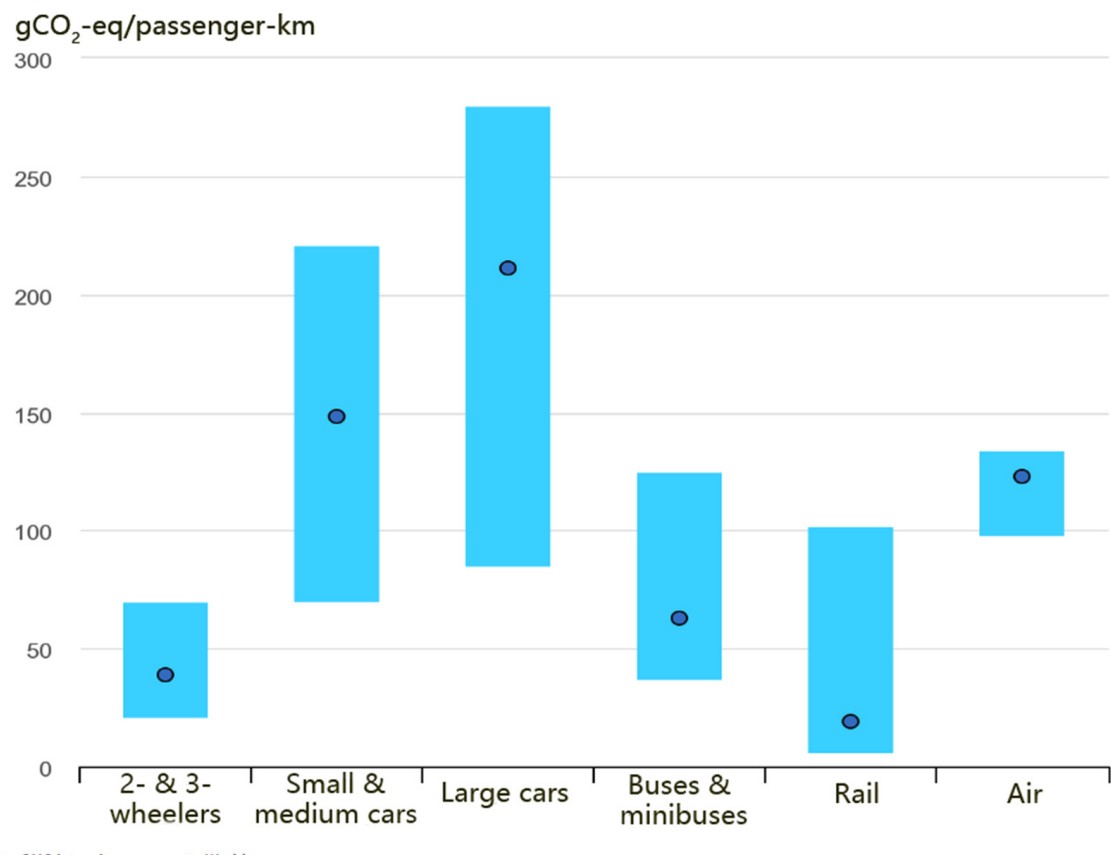

**Figure 1.** Well-to-wheel (wake/wing) GHG intensity of motorized passenger transport modes, IEA, Paris [3].

**Table 1.** Shares of low-carbon and zero-carbon fuels for global passenger rail activities for the 2020 and 2030 Net Zero Scenario.

| Year | Oil Proportion | Electricity and Hydrogen Proportion |
|------|----------------|-------------------------------------|
| 2020 | 27.39% | 72.61% |
| 2030 | 2.43% | 97.57% |

This study explores the pattern of clean energy development in the railway transportation system from three aspects. Firstly, it reviews the exploitation of clean energy in global railway transportation. Secondly, it collects and analyzes the distribution of renewable energy resources along China's railway tracks. Thirdly, the clean energy exploitation patterns of railway transportation are discussed in the context of the complex geographical environment of China. Furthermore, we envisage the prospect of multi-energy complementary utilization for a railway energy-exploitation model. Finally, we provide suggestions for the future development of clean energy exploitation in the railway transportation system.

## 2. Renewable Energy Exploitation in Railway Transportation

Although renewable energy technologies such as photovoltaic technology are not new, renewable energy in railway transportation is not widespread. Planners cannot systematically plan and design renewable energy integration development into rail transit in China. This section mainly introduces the background and trends of low carbon operations in railway transportation and the development and application of renewable energy technology along railway lines in order to summarize and put forward the exploitation mode of clean energy for railway transportation.

### 2.1. The Trend of Decarbonization of Railway Transportation Energy

The railway industry is competitive when it comes to applying renewable energy, as reflected in the two aspects of traction and non-traction power supplies. The diverse railway traction power supplies range from electric energy, generated from coal to hydrogen, to renewable ones. In non-traction energy consumption, railways can also achieve the large-scale exploitation of various types of energy brought by renewable energy. Contrary to the linear distribution of traction power consumption, non-traction energy consumption is relatively concentrated in the stations [4]. To sum up, low-carbon railway operation is feasible by promoting the application of renewable energy sources along the railways [5]. In addition to the primary renewable energy sources, hydrogen also plays a key role in achieving carbon neutrality as a secondary source to make up for the shortage of clean energy. The following section introduces the development trends of solar energy, wind energy, hydropower, and hydrogen energy for the railway system.

### 2.2. Photovoltaic Exploitation in Railway Transportation

The railway system has recently experienced the application of large-scale solar power-generation technology. China's railway system has started to research, experiment, and promote of new kinetic and solar energy along the railway lines, which is at an early stage. Some large-scale power stations and renovated and expanded hybrid trains have made outstanding achievements in utilizing new kinetic and solar energy technologies in recent years, such as the Qinghai-Tibet Railway and Beijing South Railway Station, and Shanghai Hongqiao. Jian et al. introduced two solar power generation applications in Shanghai, China [6]. The roof of Xiongan high-speed railway station is paved with a 42,000 m$^2$ photovoltaic power generation device, and the total installed capacity is 6 MW. The annual power generation capacity can reach 5.8 million kWh, reducing carbon dioxide production by 45 million tons annually and providing about 20% of its total demand with clean electricity. Zhiming et al. study the optimal planning of distributed photovoltaic generation (DPVG) and energy storage systems (ESSs) for the traction power supply system (TPSS) of the high-speed railway. This lecture demonstrates the potential and applicability of DPVG and ESS to the high-speed railway industry [7]. Fuwei et al. proposed a new evolutionary model of a railway energy supply system (RESS) for railway photovoltaic integrated systems (RPISs) by constructing a three-in-one "traction storage information integration station" (TSIIS) [8]. Zhujun et al. developed a GIS-based approach to assessing the photovoltaic potential along railways and on the rooftops of railway stations [9]. The Beijing-Shanghai high-speed railway (HSR) is used as a case study. Its total PV potential reaches 5.65 GW (of which the station potential accounts for 264 MW, approximately 4.68% of the total potential), with a lifelong generation capacity of 155 TWh, which corresponds to approximately 12% of the total new installed capacity for China in 2020.

There are many studies worldwide on the photovoltaic potential of railway systems. Olexandr et al. proposed hybrid photovoltaic systems with batteries for rail transport infrastructure [10]. Sorensen et al. map solar resource data to the GIS system and matched them with habitat-based demand models, including population density, and energy demand intensity, for the study of solar photovoltaic applications [11]. Hayashiya et al. describe the application capacity and development prospects of photovoltaic devices on the roof of railway station platforms [12]. Ibragim et al. considered the concept of using a distributed solar power plant, which was set up on the right-of-way side of the railroad [13]. H Kim et al. designed a method to improve the power generated by solar trains with photovoltaic panels. The PV potential of railway routes can be calculated using GIS [14]. Vasisht et al. established a mode for railway passenger cars to have photovoltaic panels on top of them [15]. Kilic et al. presented a method for installing photovoltaic panels on the roof of the M1 light metro line at Istanbul Airport to power train lighting facilities [16]. Morita et al. introduced the PV model of the Tokyo power station, including installing photovoltaic modules on the rooftop and evaluating the impact of shading on the efficiency of the surrounding buildings [17].

In conclusion, the current railway photovoltaic power generation models are only feasible in specific regions. Most photovoltaic modules installed on the top of trains or stations have a limited output. They are mostly used in train lighting systems or services and not for the energy needed to pull physical loads.

### 2.3. Wind Energy Exploitation in Railway Transportation

China has copious wind resources. Over the past decade, the cumulative installed capacity of wind energy has grown nearly six-fold. By 2021, China's cumulative installed wind energy capacity reached 268 GW, accounting for 35.4% of the global total. In terms of long-term planning, the cumulative installed capacity is expected to reach 358 GW by 2025 [18].

Wind energy resources have been extensively assessed around the world. Archer and Jacobson [19] analyzed the spatiotemporal distribution of wind resources at 80 m in the United States based on these measurement results. Ilkilic et al. [20] evaluated wind energy potential in Turkey and calculated the theoretical potential of onshore and offshore wind energy. Mentis et al. [21] studied wind energy potential at 80 m in Africa. The results show that the technical potential of wind energy exceeds electricity consumption in most regions. Hernandez et al. [22] analyzed the wind energy resources in northern Mexico, where the wind speed shows an obvious diurnal variation of a significant increase during the nights; more precisely, between 16:00 and 06:00.

Wind energy exploitation for highway traffic can provide a reference for the decarbonization of railways. For example, the State Highway Corporation of Israel tried to install small- and medium-sized wind turbines on poles of streetlights to take advantage of the sea breeze along the Mediterranean coastline. At present, the application of wind energy for railway systems mainly powers the monitoring equipment and other infrastructure in remote areas [23,24]. In addition, the installation of small vertical axis wind turbines in the tunnel can convert the wind energy generated by the train operation into electricity [25]. However, installing such wind power facilities is costly, and the power generation capacity is difficult to predict. For example, if no train passes, the wind turbines cannot generate electricity, resulting in inefficiency and higher system costs [26]. In addition to grid-connected wind turbines, several developed countries have researched the possibility of installing wind turbines in transport spaces [27]. Further research is needed to consume wind energy along railway lines extensively.

### 2.4. Hydropower Exploitation in Railway Transportation

The principle of hydroelectric power generation is to rotate a turbine through falling water, which drives the generator. Through a series of processes, the potential energy of water is converted into mechanical and electrical energy. The reserves of hydroelectric resources are closely related to factors such as the water volume and water level of rivers, resulting in various resource abundance in different regions. In this context, it is necessary to evaluate the potential of hydropower resources in different regions.

The hydropower energy potential is usually divided into (a) total theoretical potential, (b) technical potential, and (c) economic feasibility potential. The total theoretical potential indicates that the total electricity may be generated if all available water resources are used for this purpose. The technically exploitable potential of a system represents existing technologies' attractiveness and readily available hydropower capacity. The economically feasible potential is the hydropower capacity that can be built up after the feasibility study of each site at the current price and if positive results are generated.

In reference [28], Beatrie Wangner et al. summarize the history of hydropower development in Austria and emphasize the importance of hydropower resources for energy structures. They also describe the political, economic, and environmental challenges that hydropower would face in the future. In reference [29], Sharma et al. elaborate on the significance of hydropower development in Nepal and put forward the problems that must be considered in hydropower development. OAC Hoes et al. formally assessed the

hydropower potential at multiple locations based on the slope and flow of each river in the world [30]. Zimmy et al. analyzed the exploitation of hydropower resources across the continents [31]. The energy structure of China is dominated by coal, accounting for 71.13% of the country's total power generation [32]. Hydropower is second only to coal, accounting for 14.6%. Therefore, hydropower research in China has also attracted the attention of many experts and scholars, such as the authors of [33,34]. In reference [35], the authors describe the development of hydropower in China in the past 40 years and expound on the deterrent of hydropower development in China. To achieve carbon neutrality by 2060, the vigorous development of hydropower is indispensable.

Some countries have already launched projects to utilize hydropower for railways. The Rhaedia railway in Switzerland has been using water energy to power its passenger trains Since 2013 [36]. Austria's Kibu has built a hydro-ecological power station dedicated to supplying electricity for railway traction, with two under construction [37]. Norwegian trains have used emission-free renewable energy from hydropower since 2022, making Norway a leader in green rail transport [38].

The above research and projects conclude that the exploitation of hydropower for railways depends on the local hydropower station. In Section 3.3, the study will introduce the distribution of hydropower resources along the railways of China.

### 2.5. Hydrogen Exploitation in Railway Transportation

Due to their fluctuating nature, renewable energy sources cannot continuously and stably output electricity for a long time, resulting in mass wind and solar abandonment. Therefore, energy storage technology is indispensable. Energy storage technology can ensure stable power output and improve the ability of the power grid to absorb intermittent renewable energy [39].

As a new energy source, hydrogen energy has high energy density, large energy storage scalability, and low energy-capacity costs. It can be used as an optimal solution for long-term energy storage or seasonal energy storage, thereby effectively improving energy utilization. H Liu et al. proposed a facility planning model under the conditions of energy transfer and natural endowments [40]. Siwiec J and Xiangyu Meng et al. discussed the development strategy, technology, and industrialization status of the hydrogen energy industry for the Chinese transportation field [41,42]. This introduces the application of hydrogen-fuel cells in railways in detail. William et al. analyzed the feasibility of hydrogen-fueled trains [43]. Gino et al. confirmed that hydrogen-based vehicle traction has reached a mature technological level to replace the more polluting diesel engines [44]. Adopting this technology can also alleviate the carbon footprint of railway trains operating on non-electrified lines.

Research and testing have been carried out on the application of hydrogen fuel cells for railways [45–49] in recent years. Fuel cells (FCs) represent a promising solution for powering electric traction motors; these electrochemical devices can generate electricity directly by combining hydrogen and oxygen through catalysts. This method produces only small amounts of clean, warm water and steam, resulting in zero carbon emissions and zero pollution. Locomotives using FC power plants have been tested on a railway line in Chengdu, China, since 2012 [50]. Hydrogen-powered trams equipped with electric drives [51] are an early example of FC urban applications. Detailed numerical simulations of key components from hydrogen equipment and FC hybrid trains have been carried out on a 140 km regional line in southern Italy. This confirmed the great potential of hydrogen technology for heavy-duty transportation [52]. Sebastian et al. adopted a GIS approach based on a site-level cost model to assess the potential of wind-based hydrogen in adjacent regional orbits [53]. In northern Germany, two hydrogen-powered trains are currently operating in northern Germany [54]. Kathryn et al. forecast the emissions of hydrogen-powered trains in the UK and Japan and point out the direction that the two countries need to work towards to achieve the goals of the Paris Agreement in the future [55].

Although hydrogen is a secondary energy, it plays an important role in railway multi-energy complementation. Reviewing the application of hydrogen energy in railways can contribute to the cleanliness of railway energy in China.

## 3. Renewable Energy Potential Distribution along Railway Lines

Exploring the distribution of renewable energy along the railway is the premise of the actual exploitation of renewable energy. This section describes the distribution of solar, wind, and hydropower combined with the railway network.

### 3.1. Distribution of Solar Energy Resources along the Railway

Solar energy possesses the advantages of abundance and wide distribution. China's average annual solar radiation is about $1 \times 10^{16}$ kWh, equivalent to $1.2 \times 10^{12}$ tons of standard coal.

Solar energy resources in China can be divided into four classes according to latitude, longitude, and climate [56]. According to the traffic data of the geographic data platform of Peking University, we show the overlay distribution of railway lines and solar energy resources in China in Figure 2. Figure 2 depicts the geographical distribution of these four class areas in China. As can be seen from the legend in Figure 2, the annual radiation dosage is greater than 1860 kWh/m$^2$ in the Class I area, 1500–1860 kWh/m$^2$ in the Class II area, 1200–1500 kWh/m$^2$ in the Class III area, and less than 1200 kWh/m$^2$ in the Class IV area. We can conclude the main distribution characteristics of solar energy resources in China from Figure 2. The annual total radiation in the western region is higher than in the eastern region. Except for Tibet and Xinjiang, the radiation in the south is generally lower than in the north. The radiation is highest in the Tibetan Plateau and lowest in the Sichuan Basin [57].

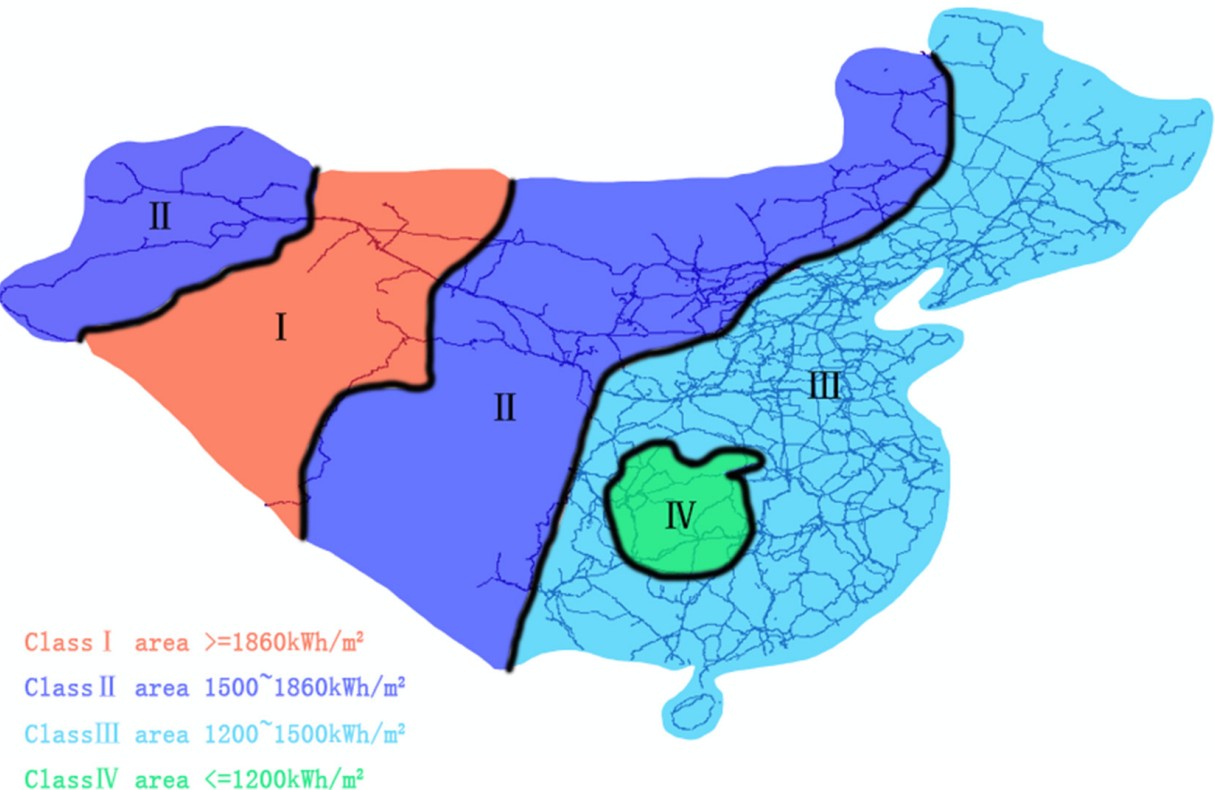

**Figure 2.** Distribution of railway lines and solar resources in China.

The length of electrified and non-electrified railways varies across the four solar energy resource regions. The lengths of electrified railways and non-electrified railways in the area I are 1075 km and 1894 km, respectively; in the area II, they are 31,489 km and 12,814 km;

in the area III, they are 57,182 km and 20,179 km, respectively; in the area IV, the lengths are 12,331 km and 1860 km, respectively [58]. According to Figure 2, the rich solar energy resources would benefit the future development of electrified railways in areas I and II.

Section 2.2 introduces photovoltaic exploitation for railway transportation. This study explores the integration potential of the railway and photovoltaic energy.

The integration of railways and photovoltaics is divided into three types of processes: the integration of carriers and photovoltaics, the integration of infrastructure and photovoltaics, and the integration of service facilities and photovoltaics. The integration of carrier and photovoltaic refers to installing photovoltaic modules on the top of the train compartment. Its most suitable application fields are non-electric railway rolling stocks. Integrating infrastructure and photovoltaic refers to installing photovoltaic modules along the railway line. The integration of service facilities and photovoltaics refers to the configuration of photovoltaic modules on station roofs. The formula for calculating the photovoltaic output potential of different integration modes is as follows [59]:

The photovoltaic output of railway carriers:

$$P_t = n_t \cdot S_t \cdot R_a \cdot \rho, \tag{1}$$

where $P_t$ is the photovoltaic output of the carrier, $n_t$ is the number of trains, $S_t$ is the photovoltaic area that can be laid on the train's roof, $R_a$ is the national average annual radiation, and $\rho$ is the conversion efficiency of the photovoltaic modules.

The photovoltaic output of railway infrastructure:

$$P_r = (l_1 \cdot c \cdot R_1 + l_2 \cdot c \cdot R_2 + l_3 \cdot c \cdot R_3 + l_4 \cdot c \cdot R_4) \cdot \rho, \tag{2}$$

where $P_r$ is the photovoltaic output of the carrier, $l_1$, $l_2$, $l_3$, $l_4$ are the lengths of photovoltaic lines that can be laid in the radiation areas of Class I, Class II, Class III, and Class IV, respectively. $c$ is the average PV module width, and $R_1$, $R_2$, $R_3$, and $R_4$ are the average annual radiation in the areas of Class I, Class II, Class III, and Class IV, respectively.

The photovoltaic output of railway service facilities:

$$P_s = (n_{s1} \cdot R_1 + n_{s2} \cdot R_2 + n_{s3} \cdot R_3 + n_{s4} \cdot R_4) \cdot S_a \cdot \rho, \tag{3}$$

where $P_s$ is the photovoltaic output of railway service facilities; $n_{s1}$, $n_{s2}$, $n_{s3}$, and $n_{s4}$ are the number of stations in the radiation areas of Class I, Class II, Class III, and Class IV, respectively, and $S_a$ is the average station area.

If the growth rate of the number of trains, from 2021 to 2025, is consistent with the scale of the railway network, the space utilization rate (of the trains) that can be installed with photovoltaics will reach 10%, with other conditions remaining unchanged. The annual photovoltaic output of China's rail transit vehicles in 2025 will be $0.11 \times 10^8$ kWh; it will reach $0.27 \times 10^8$ kWh in 2030 and $0.44 \times 10^8$ kWh in 2035.

If the scale of the railway network grows in the same way for the four types of light radiation areas, the photovoltaic space utilization rate that can be laid on the line will reach 10%. The annual photovoltaic output of China's rail transit infrastructure could reach $4.18 \times 10^{10}$ kWh in 2025. It will reach $1 \times 10^{11}$ kWh by 2030, and $1.66 \times 10^{11}$ kWh by 2035.

Suppose the growth of the number of railway stations in the four types of light radiation areas is consistent with the growth in the scale of the railway. In that case, the utilization rate of photovoltaic space which can be laid in the station is 10%, with other conditions remaining unchanged. The annual output of photovoltaics in China's railway stations could reach $2.77 \times 10^8$ kWh by 2025, $6.62 \times 10^8$ kWh by 2030, and $1.09 \times 10^9$ kWh by 2035.

### 3.2. Distribution of Wind Energy Resources along the Railway

China's regional classification of wind energy resources is mainly based on the regional high-efficiency wind energy density and the sustainable cumulative hours throughout the

year. Areas with an annual mean effective wind energy density greater than 200 W/m² and an annual cumulative hour of 3–20 m/s wind speed of greater than 5000 h are classified as wind energy-rich areas. Those areas with an annual average effective wind energy density of 150–200 W/m² and an annual cumulative hour of 3–20 m/s wind speed between 3000 and 5000 h are classified as areas with abundant wind energy. Areas where the annual average effective wind energy density is 50–150 W/m², with an annual cumulative hour wind speed of 3–20 m/s of between 2000–3000 h, are classified as wind energy-available areas. An area where the annual mean effective wind energy density is below 50 W/m² and the annual cumulative hour of wind speed is 3–20 m/s of below 2000 h is classified as a wind energy-poor area [60].

The whole country can be divided into four classes according to the status of wind energy resources and construction costs in different regions. The overlay distribution of railway lines and wind energy resources in China is shown in Figure 3.

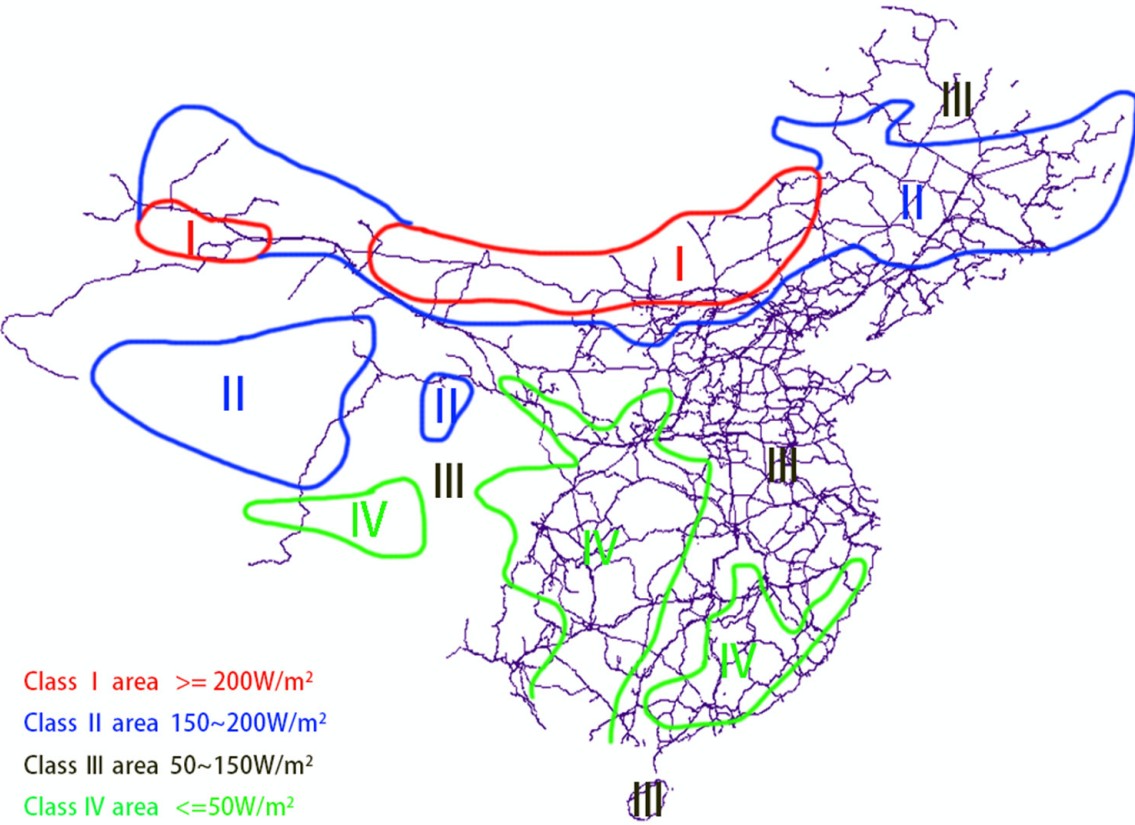

**Figure 3.** Railway lines and wind resources distributed in China.

According to Figure 3, railway wind power should focus on the western region, which has sufficient wind energy resources, such as Qinghai-Tibet Railway, and the northern region, such as Lanxin Line, Ganquan County.

### 3.3. Distribution of Hydropower Resources along the Railway

China has abundant hydropower resources but an unbalanced distribution of it. Figure 4 shows the relevant statistical data for hydropower resources in various regions of China [61,62].

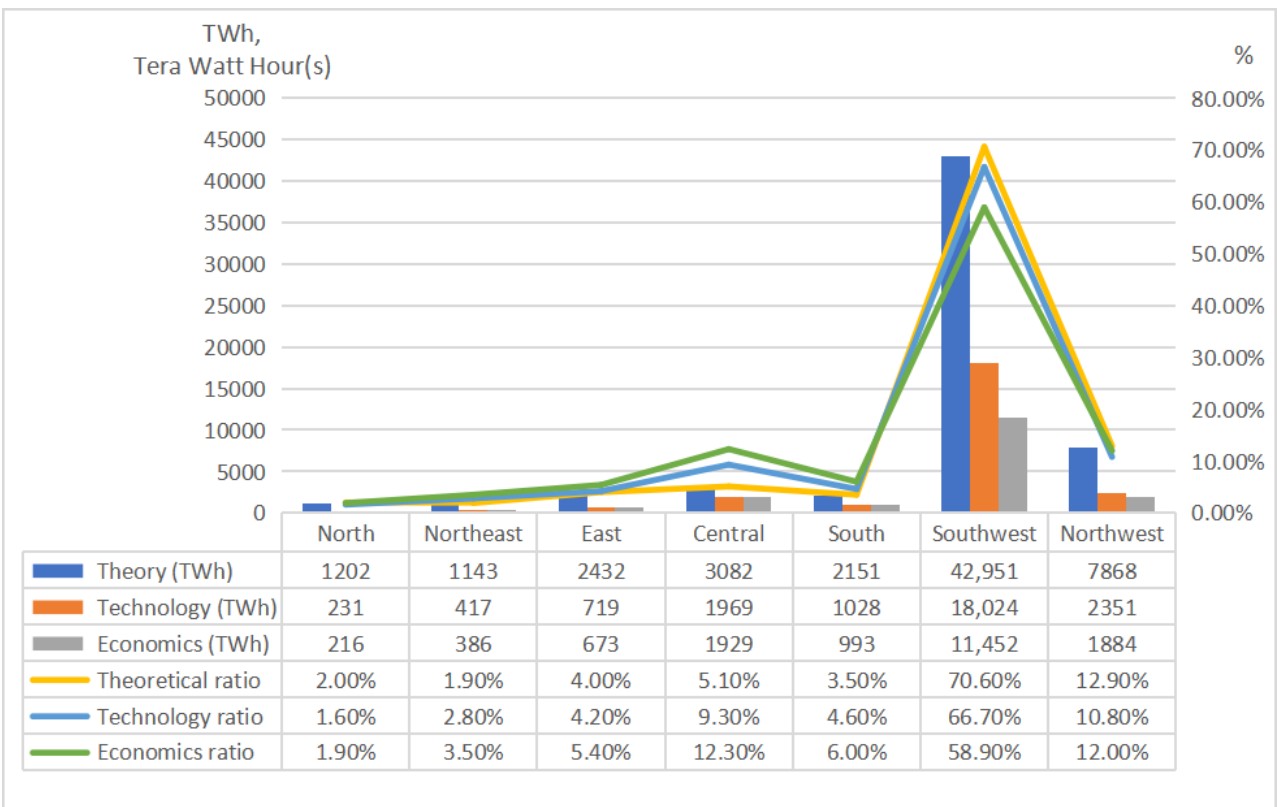

**Figure 4.** Distribution of water resources in China.

According to Figure 4, the theoretical reserves of hydropower resources in different regions of China vary greatly. Because the development of hydropower resources is limited by terrain, technological achievement, economic level, and other conditions, the degree of technical and economic development reserves in different regions of China varies. The theoretical reserves in southwest China are more than twice the sum of the reserves in other regions. The technical and economic ratios are slightly lower than the theoretical reserves. However, they also account for 66.7% and 58.9% of the national total. Accordingly, prioritizing the development of technology or the economy in the southwest region can give full play to its outstanding advantages in hydropower resources and supply this to other regions. Although the theoretical reserves of hydropower resources in the northwest region rank second, it is only 12.9%, less than one-fifth of the theoretical reserves in the southwest region. Although the theoretical reserves of hydropower resources in central and southern China are lower than those in the northwest region, the ratio of their technical and economic power generation to theoretical reserves is higher than that in northwest region. Therefore, the central and southern regions have a higher level of technological and economic development.

China's hydropower development reached a new height after 1950. The ratio of annual power generation to scientific and technological input can be used as an important index to evaluate the exploitation of hydropower resources [63]. Table 2 shows the hydropower development of major rivers in China, where the data are from the Ministry of Water Resources of the people's Republic of China (Available online: http://xxfb.mwr.cn/slbk/slfd/zgsdjs/ (accessed on 3 March 2022)).

China's hydropower resources are concentrated in the Sichuan, Yunnan, and Tibet regions, accounting for about two-thirds of the total sum hydropower supply. Sun et al. introduced the main distribution of hydropower stations in Yaan Sichuan, and parts of Tibet are ideal for hydropower station construction [64]. These areas have great railway transport energy potential.

**Table 2.** Hydropower development of each river (Development level = Generated Energy of Technical exploitation amount/Generated Energy of Developed Quantity. For example, Yangtze River Basin's Development level = 2925/11,879 = 24.62%).

| Basin | Technical Exploitation Amount | | Developed Quantity | | Development Level (%) |
|---|---|---|---|---|---|
| | Installed Capacity (MkW) | Generated Energy (TWh) | Installed Capacity (MkW) | Generated Energy (TWh) | |
| Yangtze River Basin | 25,627 | 11,879 | 6973 | 2925 | 24.62 |
| Yellow River Basin | 3734 | 1361 | 1203 | 465 | 34.17 |
| Pearl River Basin | 3129 | 1354 | 1810 | 786 | 58.05 |
| Haihe River Basin | 203 | 48 | 80 | 20 | 41.67 |
| Huaihe River Basin | 66 | 19 | 31 | 10 | 52.63 |
| in North China | 1682 | 465 | 640 | 152 | 32.69 |
| Southeast coast of the river | 1907 | 593 | 1165 | 363 | 61.21 |
| Southwest river | 7501 | 3732 | 932 | 443 | 11.87 |
| Inland river and XinJiang River | 1847 | 806 | 229 | 85 | 10.55 |
| Total | 54,164 | 24,740 | 13,098 | 5259 | 21.26 |

The following section will combine specific regions to introduce examples of renewable energy selection depending on local conditions and the application prospects of renewable energy in the multi-energy complementary energy exploitation mode.

*3.4. The Public Renewable Energy Distribution Dataset*

Research on regional energy distribution requires a large amount of data to support it. There are many energy databases currently available for researchers. Table 3 briefly introduces the publicly available databases mostly used in energy distribution research.

**Table 3.** Energy dataset.

| Dataset | Objects | URL |
|---|---|---|
| NASA | solar energy | https://power.larc.nasa.gov/ (accessed on 3 March 2022) |
| METEONORM | solar energy | https://meteonorm.com (accessed on 3 March 2022) |
| SolarGIS | solar energy | http://solargis.info (accessed on 3 March 2022) |
| DLR ISIS | solar energy | http://www.pa.op.dlr.de/ISIS (accessed on 3 March 2022) |
| SWERA | solar energy | http://maps.nrel.gov/swera (accessed on 3 March 2022) |
| 3TIER Solar Time Series | solar energy | www.3tier.com/products (accessed on 3 March 2022) |
| NCEP/NCAR | solar energy | http://rda.ucar.edu/datasets/ds090.0 (accessed on 3 March 2022) |
| MERRA-2 | wind energy | https://gmao.gsfc.nasa.gov/reanalysis/MERRA-2 (accessed on 3 March 2022) |
| National Meteorological Science Data Center | wind energy | http://data.cma.cn (accessed on 3 March 2022) |
| ERA5 reanalysis dataset | wind energy | https://cds.climate.copernicus.eu/cdsapp#!/search?type=dataset (accessed on 3 March 2022) |
| MODIS | wind energy | https://ladsweb.modaps.eosdis.nasa.gov (accessed on 3 March 2022) |
| National Bureau of Statistics—China Statistical Yearbook | hydropower | www.stats.gov.cn/tjsj/ndsj (accessed on 3 March 2022) |
| China Electricity Council | hydropower | www.cec.org.cn (accessed on 3 March 2022) |
| bp world energy statistics report | integrated energy | https://www.bp.com (accessed on 3 March 2022) |
| IEA | integrated energy | https://www.iea.org/fuels-and-technologies/renewables (accessed on 3 March 2022) |
| Energy Basic Database | integrated energy | http://db.energy.ckcest.cn (accessed on 3 March 2022) |
| NERL | integrated energy | https://www.nrel.gov/research/data-tools.html (accessed on 3 March 2022) |

## 4. China's Clean Energy Exploitation Model for Railway Transportation Systems

*4.1. Renewable Energy Exploitation Patterns According to the Geographical Area*

Renewable energy is friendly to the environment and has a large power generation potential, yet this supply is restricted by various factors, such as geographical environment,

climatic conditions, resource richness, and technology maturity. Therefore, it is necessary to thoroughly explore the distribution of clean energy resources and comprehensively evaluate the availability, efficiency, economy, and stability of renewable energy supply patterns.

China's natural geography presents a pattern of "three steps of terrain plus three natural areas", as shown in Figure 5 (Map data from Google Earth).

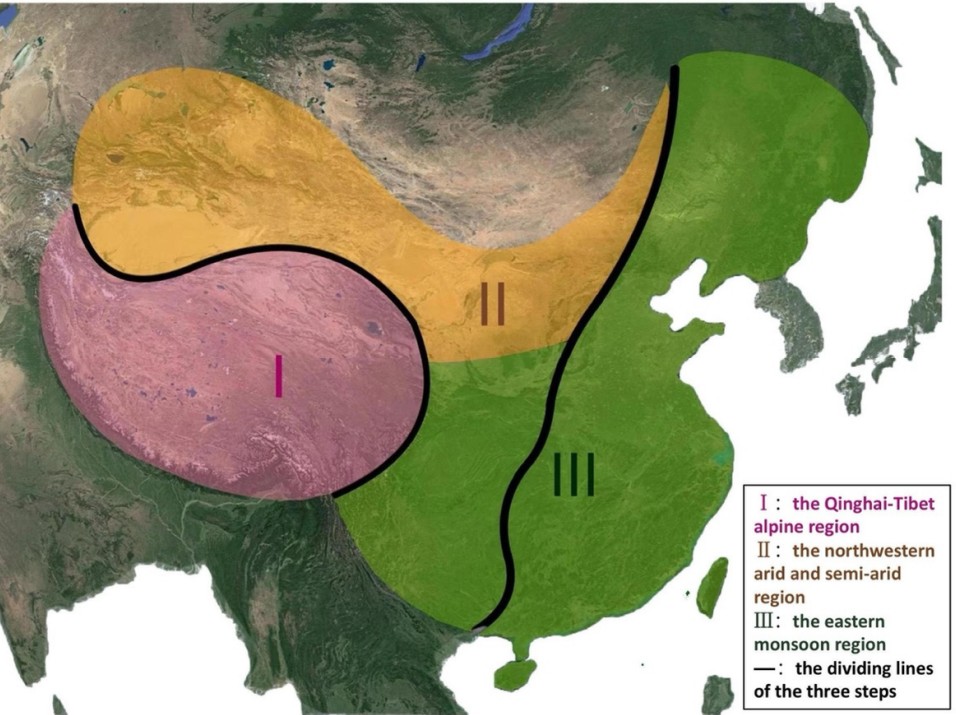

**Figure 5.** Abridged general view of the "three steps of the terrain plus three natural areas".

China's terrain gradually descends in three steps from west to east. The black bold lines show the dividing lines of the three steps in Figure 5. The first step includes the Qinghai-Tibet Plateau and the Qaidam Basin, with an average altitude of more than 4000 m, which is bounded by the Kunlun Mountains, Altun Mountains, and Qilian Mountains in the north and the Hengduan Mountains in the east. In the second step, plateaus and basins are intertwined, including the three major plateaus and three basins in northern and central China, with an average altitude of between 1000–2000 m. The third step in the east has many plains and hills, with an average altitude of less than 500 m. The Greater Khingan, Taihang Mountains, Wushan Mountain, and Xuefeng Mountain separate the second and third steps [65].

Besides the three steps of terrain, China is divided into three natural regions according to differences in natural geographical elements, such as temperature and precipitation. These three natural regions are the Qinghai-Tibet alpine region, the northwestern arid and semi-arid region, and the eastern monsoon region [66], as shown by the pink, yellow, and green regions in Figure 5, respectively. The Qinghai-Tibet alpine region is mainly located on the first step of the terrain. The three mountain ranges, such as the Kunlun Mountains mentioned above, are also the boundary between the Qinghai-Tibet alpine region and the northwestern arid and semi-arid regions. In the eastern part of the Qinghai-Tibet alpine region, the 3000 m contour line naturally demarcates it from the eastern monsoon region. According to the precipitation, the northwestern arid and semi-arid region and the eastern monsoon region are bounded by the 400 mm isohyet, extending from the Greater Khingan Mountains to the eastern edge of the Qinghai-Tibet Plateau.

The macroscopic pattern of the natural structure of "three natural areas plus three steps of terrain" determines their respective clean energy exploitation modes. Plateaus,

plains, and basins are interlaced with mountains, rivers, and hills. Clean energy resources formed by the natural geographical features have created unique conditions for railway transportation. For instance, Ningxia, located on the second step, with a high altitude and thin air, has an annual average total solar radiation of about 6680–8400 MJ/m$^2$ and annual sunshine hours totaling 3000 h, making it effective land for solar energy exploitation. Besides, Ningxia is in the strong wind belt of Gansu, Inner Mongolia, and Liaoning and has large-scale wastelands for wind power generation [67]. This abundant solar and wind energy can provide considerable electricity for many railway lines, such as the Baoyin high-speed railway and the Baolan railway in Ningxia.

Tibet is in the first step of the "three steps of terrain" and the Qinghai-Tibet alpine region of the "three natural areas" in China. The unique geographical environment and climatic conditions make it rich in clean energy resources [68]. Section 4.2 will select Tibet as a typical area to briefly discuss the exploration modes of clean energy in this region's railway transportation system.

*4.2. Exploration of Energy Transportation Integration in Tibet*

The construction of railways in Tibet has evidenced breakthroughs with the Qinghai-Tibet railway and the Sichuan-Tibet railway completed and operational. The Qinghai-Tibet railway traverses the Kunlun Mountains, Hoh Xil Mountains, Tanggula Mountains, Tuotuo River, and Tongtian River along the way. The Sichuan-Tibet railway, with a total length of 1932.9 km, spans the Western Sichuan Plateau and the Hengduan Mountains, where the Minjiang, Yalong, Jinsha, Lancang, Nu, and Yarlung Zangbo rivers intersect. There are abundant renewable energy sources along the two railway lines.

The special geographical environment of Tibet is a double-edged sword for the integrated development of energy and transportation. On the one hand, the geographical advantages of low latitude, high altitude, and thin air make this region the most abundant solar energy resource in the country; on the other hand, permafrost, alpine hypoxia, and ecological environmental protection are also challenges that must be faced in the development and consumption of clean energy. The fragile geographical environment and it's vulnerabilities make it necessary for Tibet to adapt to local conditions and develop collaboratively with multiple energy sources. Only then can we give full play to the advantages of the energy exploitation model, protect the fragile ecological environment along the route, and form a green, low-carbon, and long-term development mechanism.

Li Yunong constructed a comprehensive evaluation index system for renewable energy supply modes in small- and medium-sized stations (of the Sichuan-Tibet railway) based on the order relational analysis method, namely the G1 method and the improved data envelopment analysis (DEA) model, which uses the efficiency index to measure energy consumption patterns effectively [69]. It is concluded that solar energy is the most suitable energy consumption mode, which is consistent with the actual construction situation. According to the discussion of Cai Guotian et al. on Tibet's energy situation [70], we propose the multi-energy complementation exploitation model along the Qinghai-Tibet railway and the Sichuan-Tibet railway, with the following three assumptions. First, priority should be given to developing small hydropower sources, while considering biogas and solar energy for ecological environments, such as Nyingchi and Qamdo, which pass through the Yalin section of the Sichuan-Tibet line. Second, in areas adjacent to "one river and two rivers" in the Lalin section of the Sichuan-Tibet line, solar energy and small hydropower are planned, and biogas is developed simultaneously. Third, in the relatively arid Ali area and some high-altitude areas with "one river and two rivers" that the Qinghai-Tibet line passes through, the development of solar energy and small hydropower is equally important. Biogas is moderately developed locally within a certain range, establishing the overall layout of multi-energy complementary, integrated, and coordinated development. Multi-energy complementation maximizes the organic integration of clean energy. It improves the system's stability, reliability, and safety, along with the absorption capacity of renewable energy and the system's overall energy efficiency.

The complex geographical environment of Tibet makes the choice of regional energy development mode critical. With the development of science and technology and the support of national policies in recent years [71], the exploration and development of clean energy in Tibet has entered a stage of steady progress. During the construction of railways in Tibet, clean energy, such as solar energy, would create new value in the future.

*4.3. Enlightenment to Clean Energy Exploitation Model for Railway Transportation System*

In summary, according to regional railway energy demand and resource endowment characteristics, it is indispensable to comprehensively consider the characteristics of different renewable energy supplies and local conditions. The main factors considered in developing and consuming regional energy include climate and hydrology, topography, distance from the railway network, construction and maintenance costs, technological maturity, etc. To achieve self-consistency in renewable energy for the railway transportation system, we should plan regionally differentiated scenarios and coordinate multiple renewable energy resources along the railway.

## 5. Conclusions

This study introduces the background and significance of clean energy development in railway transportation, and the trend of clean energy exploitation for railways. This study reviews the distribution of renewable energy along the railway lines of China and shows the application prospects for multi-energy complementarity for energy exploitation in railway transportation. From the established projects and theoretical review, the following conclusions can be drawn:

(1) Most research focuses on exploiting photovoltaics energy along the railway, whereas few researchers have explored wind energy and hydropower for the transportation system;

(2) Due to the limitations of geography, technology, and human factors, the clean energy resource potential along the railways has not been developed on a large scale;

(3) The photovoltaic potential along the railway is abundant. Taking Beijing-Shanghai Railway as an example, its photovoltaic potential is as high as 5.6 GW, and its lifetime power generation capacity is 155 TWh, equivalent to about 12% of China's new installed capacity in 2020;

(4) China's western region has vast land, abundant solar energy resources, and a low electrification degree. Further research is needed on the photovoltaic potential of non-electrified railway sections in the western region;

(5) Considering the intermittent nature of renewable energy, it is significant to introduce hydrogen energy as energy storage.

In general, the importance of exploiting renewable energy for powering the railway system is beyond doubt. However, the studies and applications of railway renewable energy are still in their infancy, requiring further research. Regional energy development needs to consider the uncertainty caused by the probability distribution characteristics of renewable energy on different space-time scales. Exploiting a multi-energy complementary supply and energy storage system could efficiently cope with the randomness of renewable energy supplies, volatility, and intermittency. Furthermore, the layout of the clean energy system should be planned in combination with external factors, such as policies and economics.

**Author Contributions:** Conceptualization, J.T., R.S., L.L. and Y.J.; methodology, L.L. and Y.J.; data curation, L.L. and Y.J.; writing—original draft preparation, L.L. and Y.J.; writing—review and editing, J.T. and R.S.; supervision, J.T. and R.S.; project administration, J.T. and R.S.; funding acquisition, J.T. and R.S. All authors have read and agreed to the published version of the manuscript.

**Funding:** This research is financially supported by the National Key R&D Plan Foundation of China (Grant No. 2021YFB2601300) and the Fundamental Research Funds for the Central Universities under Grant No. 2020MS017.

**Institutional Review Board Statement:** Not applicable.

**Informed Consent Statement:** Not applicable.

**Data Availability Statement:** Not applicable.

**Acknowledgments:** Acknowledgement for the data support from "Geographic Data Sharing Infrastructure, College of Urban and Environmental Science, Peking University (http://geodata.pku.edu.cn (accessed on 25 April 2022))".

**Conflicts of Interest:** The authors declare no conflict of interest.

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
