# Peer review of "A Review of Clean Energy Exploitation for Railway Transportation Systems and Its Enlightenment to China"

_sustainability, doi:10.3390/su141710740_

Round 1
Reviewer 1 Report
* The word "utilization" isn´t correct. Change it to "consume". "Use" and "consume" words aren´t the same.
* Line 57: Don´t use "This paper", use "This study/analyses/assessment".
* Line 66: "State of the art".
* Avoid writing: Reference |x| designs/takes/study, etc. Rewrite all these kinds of sentences. (example: line 136/138/139)
* The quality and size of Figure 2 are too low, improve it.
* The wording of the document must be reviewed by a native English speaker
Author Response
Dear Reviewer:
We deeply appreciate for your constructive comments. They greatly helped us to improve the manuscript. Several concerns were raised. Below are our responses to each comment, with corresponding changes made to the revised manuscript.
* The word "utilization" isn´t correct. Change it to "consume". "Use" and "consume" words aren´t the same.
We agree with the reviewer that the word "utilization" isn´t correct. Our study focuses on the power supply potential exploitation of clean energy resources in the railway transportation field. We have revised our manuscript accordingly. The term “Clean Energy Utilization” in the title has been changed to “Clean Energy Exploitation”. Besides, we have thoroughly revised the manuscript and marked corresponding changes in this submission.
* Line 57: Don´t use "This paper", use "This study/analyses/assessment".
This comment is well noted, and we have revised the manuscript accordingly. The modifications are marked up using the “Track Changes” function in MS Word.
* Line 66: "State of the art".
We agree with the reviewer that the term “State of the art” is not precise enough. We have revised the title of section 2 to “Renewable energy exploitation in railway transportation” and other corresponding descriptions in the manuscript.
* Avoid writing: Reference |x| designs/takes/study, etc. Rewrite all these kinds of sentences. (example: line 136/138/139)
We appreciate this reviewer’s constructive comments. We have modified all these sentences on page 3, line 93, 115; page 4, line 120, 121; page 5, line 205, 206, 210; page 6, line 220, 224; page 11, line 353.
* The quality and size of Figure 2 are too low, improve it.
Thanks so much for the comments. We have improved the quality and size of Fig. 2.
* The wording of the document must be reviewed by a native English speaker.
We highly appreciate the reviewer’s constructive suggestion. We have re-written the manuscript. In addition, this manuscript has received professional English editing service.

Reviewer 2 Report
The review paper is indeed of interest, i have carefully reviewed the manuscript and my comments as follows:
1. In general the topic is of interest, but reader may realize that its conventional. Listing the renewable energy sources in section 2 (which is obvious sources) is fine, but its not well linked to each others, and the sections titles refer to "state of the art" but PV, wind energy...etc are not new, so what's the significance that makes it state of the art? How about hybrid the renewable sources for railway applications, and it can be used in China?
2. The abstract need to provide critical quantifiable findings.
3. in section 2.2, please be specific in the section title and use PV instead of generally refer to solar only.
4. In most tables, the numbers provided showed two decimals, for example in table 4 for Yangtze River Basin, generated energy is 11878.99 which can be rounded to 11879 , is it possible to round the numbers in all tables?
5. The Please mention in Figure 4 what the Y axis represents.
6. Figure 3 does not provide any information, I suggest to remove it or provide information related to the regions in fig 3 map.
7. in 4.1, please provide subsections based on regions in China.
8. In section 4.2, there is almost no quantifiable findings, the section also did not provide specific findings and its too general.
9. Future work is crucial in review papers, which appears to be mention in conclusion only, why, you need to provide critical recommendation of future work on research as well as recommendation of hybriding the renewable energy sources based on regions in China.
10. It is well known that China investment and research on renewable energy is enormous, but most of the 2022 references listed came from public domain reports, please include recent research published in reputable journals in 2022.
11. Please provide quantifiable results/findings in conclusion, the conclusion in its current format is just a text.
12. Please perform careful proofread of your manuscript, there are many grammatical errors and sentences are not linked.
12. One minor comment, please use proper citation, for example Professor Jia Limin et al., it should be Limin et al.
Author Response
Dear Reviewer:
We deeply appreciate for your constructive comments. They greatly helped us to improve the manuscript. Several concerns were raised. Below are our responses to each comment, with corresponding changes made to the revised manuscript.
- In general the topic is of interest, but reader may realize that its conventional. Listing the renewable energy sources in section 2 (which is obvious sources) is fine, but its not well linked to each others, and the sections titles refer to "state of the art" but PV, wind energy...etc. are not new, so what's the significance that makes it state of the art? How about hybrid the renewable sources for railway applications, and it can be used in China?
We agree with the reviewer that Photovoltaic and wind power are not new. Our innovation lies in exploitation of Photovoltaic, wind power, and other clean energy resources in the railway transportation field. Application of mixed renewable energy resources in the transportation field of China is still in the stage of theoretical research. We have modified the description in section 2 on page 3, line 66-72. Besides, we have supplemented relevant literature in section 2.2 on page 3, line 87-110 and in section 4.1.
- The abstract need to provide critical quantifiable findings.
This comment is well noted. We have revised the abstract and added critical quantifiable findings on page1, line 14-16.
- in section 2.2, please be specific in the section title and use PV instead of generally refer to solar only.
Thanks so much for the comments. We have modified the title of section 2.2 to “Photovoltaic consumption in railway transportation” and related descriptions in the manuscript.
- In most tables, the numbers provided showed two decimals, for example in table 4 for Yangtze River Basin, generated energy is 11878.99 which can be rounded to 11879, is it possible to round the numbers in all tables?
We appreciate this reviewer’s constructive comments. We have rounded the numbers in most of the tables except for the percentage ratios.
- The Please mention in Figure 4 what the Y axis represents.
We appreciate the reviewer’s careful reading. We have added the meaning of the Y axis in Figure 4, which denotes the equivalent power generation of TWH.
- Figure 3 does not provide any information, I suggest to remove it or provide information related to the regions in fig 3 map.
We highly appreciate this reviewer’s constructive comments. We have modified Fig. 3 to provide wind resources distribution information related to the railway lines in China.
- in 4.1, please provide subsections based on regions in China.
We greatly appreciate the reviewer’s careful reading of our manuscript. We have rewritten Chapter 4. In section 4.1, we introduce the influence of geographical factors on the development pattern of renewable energy in details. In section 4.2 we select Tibet as a typical area to discuss its clean energy exploitation modes in the field of railway transportation. Section 4.3 summarizes the enlightenment to clean energy exploitation model for railway transportation system
- In section 4.2, there is almost no quantifiable findings, the section also did not provide specific findings and its too general.
We agree with the reviewer that specific findings are important. We have rewritten Chapter 4. In the new section 4.2 we select Tibet as a typical area to discuss its specific clean energy exploitation modes.
- Future work is crucial in review papers, which appears to be mentioned in conclusion only, why, you need to provide critical recommendation of future work on research as well as recommendation of hybriding the renewable energy sources based on regions in China.
We agree with the reviewer that the future work is crucial for the review. We have supplemented the critical recommendation of our future work, which are “Regional energy development needs to consider the uncertainty caused by the probability distribution characteristics of renewable energy on different space-time scales. Exploiting a multi-energy complementary supply and energy storage system could efficiently cope with renewable energy’s randomness, volatility, and intermittency. Furthermore, the lay-out of the clean energy system should be planned in combination with external factors such as policies and economics.” The corresponding modifications are in this submission on page 15, line 490-497.
- It is well known that China investment and research on renewable energy is enormous, but most of the 2022 references listed came from public domain reports, please include recent research published in reputable journals in 2022.
We appreciate this reviewer’s constructive comments. We have included the recent related research papers published in 2022, which are:
[9] Chen, Z.; Jiang, M.; Qi, L.; et al. Using existing infrastructures of high-speed railways for photovoltaic electricity generation. J. Resources, Conservation and Recycling. 2022, 178, 106091.
[10] Shavolkin, O.; Shvedchykova, I.; Gerlici, J.; et al. Use of Hybrid Photovoltaic Systems with a Storage Battery for the Remote Objects of Railway Transport Infrastructure. J. Energies 2022, 15(13), 4883.
[14] Kim, H; Ku, J; et al. A new GIS-based algorithm to estimate photovoltaic potential of solar train: Case study in Gyeongbu line, Korea. J. Renewable Energy. 2022, 190, 713-729.
[57] Jia, Limin.; Cheng, Peng.; et al. Research on the development path and strategy of rail transit and energy integration under the “double carbon “target. China Engineering Science. 2022, 24 (3): 11.
They are cited on page 3, line 106, 113; page 4, line 120; page 7, line 272.
- Please provide quantifiable results/findings in conclusion, the conclusion in its current format is just a text.
We appreciate this reviewer’s constructive comments. We have added quantifiable findings in conclusion, which are:
1) Most of the research focused on the exploitation of photovoltaics energy along the railway, whereas few explored wind energy and hydropower in the transportation system.
2) Due to the limitations of geography, technology, and human factors, the clean energy resource potential along the railway has not been developed on a large scale.
3) The photovoltaic potential along the railway is abundant. Taking Beijing-Shanghai Railway as an example, its photovoltaic potential is as high as 5.6 GW, and its lifetime power generation capacity is 155 TWh, equivalent to about 12 % of China's new installed capacity in 2020.
4) China’s western region has a vast land, abundant solar energy resources, and low electrification degree. Further research is needed on the photovoltaic potential of non-electrified railway sections in the western region.
5) Considering the intermittent nature of renewable energy, it is significant to introduce hydrogen energy as energy storage.
The revised part is in this submission on page 15, line 477-489.
- Please perform careful proofread of your manuscript, there are many grammatical errors and sentences are not linked.
Thanks so much for the comments. We have re-written the manuscript. In addition, this manuscript has received professional English editing service.
- One minor comment, please use proper citation, for example Professor Jia Limin et al., it should be Limin et al.
This comment is well noted. We have modified the relevant sentences.

Reviewer 3 Report
Please see the attached file.

Author Response
Dear Reviewer:
We deeply appreciate for your constructive comments. They greatly helped us to improve the manuscript. Several concerns were raised. Below are our responses to each comment, with corresponding changes made to the revised manuscript.
Line 28:
It is stated: “As can be seen from Figure. 1, railway is the lowest energy consumption transportation mode.”The issue is it has the lowest average gCO2-eq./passenger-km but not the lowest range. It is believed the statement should be revised according to what the graph is exactly showing.
We appreciate the reviewer’s careful reading. We have modified the description of this sentence to “As can be seen from Figure 1, the railway has the lowest average gCO2-eq./passenger-km.” on page 1, line 30-31.
Line 30:
Make Figure 3 larger; it is somewhat challenging to read
Thanks so much for the comments. We have modified Figure 3 as the overlay distribution of railway lines and wind energy resources in China. The modified Figure 3 is on page 9.
Line 42:
Change the first column heading from Age to Year
We greatly appreciate the reviewer’s careful reading. We have changed the first column heading of Table 1 from “Age” to “Year”.
Line 79:
Insert a space after [5]
This comment is well noted. We have modified this typo on page 3, line 81.
Line 94:
Insert a space after the word Reference
Well noted. We have rewritten the sentence on page 3, line 93-94.
Line 119:
Delete a space after grids.
Well noted. We have deleted this sentence.
Line 152:
Insert a space before [25]
Well noted. We have inserted a space before [25] on page 4, line 151.
Line 154:
Insert a space after the word costs
Well noted. We have inserted a space after the word costs on page 4, line 154.
Line 177:
Insert a space before [33]
Well noted. We have inserted a space before [33] on page 5, line 183.
Line 206:
Insert a space after [45]
Well noted. We have rewritten the sentence on page 5, line 210-211. The original reference [45] now becomes reference [44].
Line 252:
It is written..455 km/ km2; it appears a value is missing in the denominator
Thanks so much for the comments. We have deleted the whole sentence due to the redundancy.
Line 265:
The text in Figure 2 is difficult to read: Also, maybe use a different color for zone III
Well noted. We have improved the quality of Figure 2 and changed the color of zone III to black.
Line 271-272:
It appears in Table 2 that zones II and III have the most concentrated electrified railways instead of zones III and IV, as stated. On the other hand, Figure 2 shows that the most concentrated electrified railways are in zones III and IV, as stated. Maybe a better statement that follows what each is showing would be helpful.
We appreciate this reviewer’s constructive comments. We have modified the description in this submission on page 5, line 254-255.
Line 276-288:
A calculation method is discussed. However, it is challenging to follow without pictures, schematics, etc. Including pictures, schematics, diagrams, etc., would be very helpful, indicating the various locations, points, etc., mentioned. This would facilitate an easier understanding of the method and the description of any relevant equations.
We highly appreciate this reviewer’s constructive comments. We have added the algorithm of railway photovoltaic potential evaluation. The rewritten section is on page 7, line 261-280; page 8, line 281-298.
Line 308:
It is stated, “from Table 3, most areas with abundant wind energy resources are in the west.” This is not easily inferred from the table. Maybe it would be helpful to include a map with locations.
We appreciate this reviewer’s constructive comments. We have modified Figure 3 to show the overlay distribution of railway lines and wind energy resources in China. We hope this graph would provide more information. Table 3 is deleted due to the redundancy. These modified parts are on page 9 of this submission.
Line 328:
What region is being talked about? “Theoretical reserves are more than twice….” It should be noted.
We are sorry for not being clear in the original statement. We have modified this sentence to “The theoretical reserves in southwest China are more than twice the sum of the reserves in other regions.” on page 10, line 331-332.
Line 329:
Delete the duplicate sentence “The theoretical reserves are more than twice the sum…”
Well noted. We have deleted the duplicate sentence.
Line 331:
A period is missing after the word reserves
Well noted. We have modified this sentence on page 10, line 340.
Line 336-337:
It is noted, “The technological and economic development of the central and southern China is higher than the northwest China,” but this is challenging to see from Figure 4. A better way is needed to show this or write a different statement that can be inferred easily from the figure.
We appreciate this reviewer’s constructive comments. Although the theoretical reserves of hydropower resources in central and southern China are lower than those in Northwest China, the ratio of technical and economic power generation divided by theoretical reserves is higher than that in Northwest China. For example, the theoretical volume in the central region is only 3082 TWh, and the economic development volume is 1929 TWh, with a ratio of 62.6%, higher than the 23.9% in the northwest region. We have improved the description of this sentence on page 10, line 338-342.
Line 344:
One example of the unit conversion to obtain the developmental level % in Table 4 may be useful.
We appreciate this reviewer’s constructive comments. We have added a description to the title of Table 3 (we deleted the original Table 3, so the original Table 4 becomes Table 3 in this submission) on page 10, which is “Development level = Generated Energy of Technical exploitation amount / Generated Energy of Developed quantity. For example, Yangtze River Basin’s Development level = 2925/11879 = 24.62%”
Line 348-349:
“Ya ‘an……”; there appears to be a need for consistent font and deletion of extra spaces
Thanks so much for the comments. We have modified “Ya ‘an” to “Yaan” and make it consistent in the whole manuscript.
Line 387:
Insert space after 400
Thanks so much for the comments. We have rewritten the Section 4.
Line 393:
Insert space after photovoltaic
Thanks so much for the comments. We have rewritten the Section 4.
Line 393-395:
It would be very helpful to cite a reference showing the calculations or provide an example.
We highly appreciate this reviewer’s constructive comments. We have rewritten Section 4 in this submission.
Line 397:
Insert a space after the word power
Thanks so much for the comments. We have rewritten the Section 4.
Line 405:
Stated is: “..terrain drops of the newly..” Maybe the word should be off
Thanks so much for the comments. We have rewritten the Section 4.
Line 406:
Insert space after meters
Thanks so much for the comments. We have rewritten the Section 4.
Line 426:
Delete space after east-west
Thanks so much for the comments. We have rewritten the Section 4.
Line 522:
Delete the extra period after the word system
Thanks so much for the comments. We have rewritten the Section 4.
Line 523:
There appears to be an extra space after the word electricity
Thanks so much for the comments. We have rewritten the Section 4.
Line 531-534:
Provide a reference for the demonstration project mentioned
We highly appreciate the reviewer’s constructive suggestion. We have rewritten the Section 4.
Line 535:
In the beginning, the word energy is missing the y at the end
Thanks so much for the comments. We have rewritten the Section 4.
Line 539:
Delete the extra period after [80]
Thanks so much for the comments. We have rewritten the Section 4.
Summary
The topic is very important in worldwide efforts to meet GHG emissions targets to support the goals of the Paris Agreement, e.g., limiting the global temperature rise to 2 °C. There was a good explanation of the state-of-the-art. There was a discussion about the renewable energy potential distribution along the railway lines. The resources available for each renewable energy, solar, wind, and hydropower, were discussed. More details are needed about the calculation method for solar energy resources. Some figures do not support the text states. So, other figures, pictures, diagrams, etc., are thought to be needed in support of some claims in the text and for better understanding. It would be very helpful to provide maps of all the locations mentioned and discussed.
It is believed that some brief comments on preliminary techno-economic (TEA) and lifecycle analyses (LCA) are very important. This could highlight the potential economic improvements and reductions in carbon intensities (environmental impacts) for the various renewable energy sources in the different regions. It is believed that this is very valuable to assess the feasibility of the clean energy methods for each region, as there are differences in capacity developments and utilization conditions for each region. The information would also be useful for community engagement, which can ensure the success of such projects.
As climate change continues, there are likely to be more regions that experience droughts, such as California in the United States. How can models incorporate this undesirable development? It was not mentioned in the paper, which includes hydropower to play a major role. Is a drought situation a threat in the future for China?
We highly appreciate the reviewer’s constructive suggestion. We have added some details on the calculation of railway photovoltaic potential on page 7, line 261-280; page 8, line 281-298. Our previous work is based on decades of historical data. Although hydropower can be used in Sichuan and Tibet, the exploitation of renewable energy in these areas are still dominated by photovoltaic energy. Therefore, even under the drought situation, there are alternatives available in these areas.

Reviewer 4 Report
Dear Authors,
The study is well organized and depicts an important aspect.
My observations are:
1. The second part of the title is flawed, pl check.
2. In overall context, you may compare a global model of clean energy to the study, it will enhance the impact.
3. Figures are not clear overall.
4. Titles in Section 2 are confusing. i-e: "The stae of the arts of", please check.
5. Numerical information and statistical data is the weak point of your study, reinforce it with numerical models and data.
6. Referencing styles are flawed at a couple of points, i guess it would be automatically sorted out, anyways recheck.
7. Energy dataset are quite authentic t you may have other options to have a better comparisons. (optional)
8. For the references the number of ratio of recent studies is a bit low, please look for recent articles.
Author Response
Dear Reviewer:
We deeply appreciate for your constructive comments. They greatly helped us to improve the manuscript. Several concerns were raised. Below are our responses to each comment, with corresponding changes made to the revised manuscript.
- The second part of the title is flawed, pl check.
Thanks so much for the comments. We have revised the title to “A Review of Clean Energy Exploitation in Railway Transportation System and Its Enlightenment to China”.
- In overall context, you may compare a global model of clean energy to the study, it will enhance the impact.
We agree with the reviewer that the comparison with a global model would enhance the impact of the study. Our current study focused on the related theoretical international research and the enlightenment to China. Although there are some sporadic attempts of clean energy exploitation in railway transportation system in Europe and Asia, a global model has not been built yet. We will try to build a global model in the future work.
- Figures are not clear overall.
Well noted. We have improved the qualities of all Figures in this submission.
- Titles in Section 2 are confusing. i-e: "The state of the arts of", please check.
Thanks so much for the comments. We have revised the section titles in this submission and deleted the confusing description of “the state of the arts of”.
- Numerical information and statistical data is the weak point of your study, reinforce it with numerical models and data.
We highly appreciate the reviewer’s constructive suggestion. We have added the theory of PV potential assessment for railways on page 7, line 261-280; page 8, line 281-298. Besides, we have supplemented statistical data of the photovoltaic potential of the Beijing-Shanghai high-speed rail, on page 3, line 106-110. Furthermore, we have modified Figure 3 to provide wind resources distribution information related to the railway lines in China.
- Referencing styles are flawed at a couple of points, i guess it would be automatically sorted out, anyways recheck.
Well noted. We have modified these formats.
- Energy dataset are quite authentic t you may have other options to have a better comparison. (optional)
We appreciate this reviewer’s constructive comments. Table 4 is the relevant energy data sets we have collected at present. In the next step, we intend to use these energy data sets to evaluate the clean energy potential along the railway.
- For the references the number of ratios of recent studies is a bit low, please look for recent articles.
Thanks so much for the comments. We have added some recent studies published between 2020 and 2022, which are:
[7] Zhong, Z.; Zhang, Y.; Shen, H.; et al. Optimal planning of distributed photovoltaic generation for the traction power supply system of high-speed railway. J. Journal of Cleaner Production. 2020, 263, 121394.
[8] Ning, F.; Ji, L.; Ma, J.; et al. Research and analysis of a flexible integrated development model of railway system and photovoltaic in China. J. Renewable Energy. 2021, 175, 853-867.
[9] Chen, Z.; Jiang, M.; Qi, L.; et al. Using existing infrastructures of high-speed railways for photovoltaic electricity generation. J. Resources, Conservation and Recycling. 2022, 178, 106091.
[10] Shavolkin, O.; Shvedchykova, I.; Gerlici, J.; et al. Use of Hybrid Photovoltaic Systems with a Storage Battery for the Remote Objects of Railway Transport Infrastructure. J. Energies 2022, 15(13), 4883.
[14] Kim, H; Ku, J; et al. A new GIS-based algorithm to estimate photovoltaic potential of solar train: Case study in Gyeongbu line, Korea. J. Renewable Energy. 2022, 190, 713-729.
[57] Jia, Limin.; Cheng, Peng.; et al. Research on the development path and strategy of rail transit and energy integration under the “double carbon “target. China Engineering Science. 2022, 24 (3): 11.
[68] Mima, Chiren.; Niu, Xiaochun.; Yao, Liang.; Labaton, Pearl. Typical application and development trend of renewable energy in Tibet. Electrical age. 2020, 10, 22-24 + 35.
They are cited on page 3, line 102, 104, 106, 113; page 4, line 120; page 7, line 272; page 13, line 414.

Round 2
Reviewer 1 Report
The manuscript has been greatly improved and I consider that all the initial suggestions have been followed, in addition to the significant improvement of the English grammar.
Reviewer 2 Report
I have reviewed the revised version of the manuscript and i found the authors have provided appropriate responses and performed the required changes/revisions to the manuscript.